# Understanding the origin of lithium dendrite branching in $Li_{6.5}La_3Zr_{1.5}Ta_{0.5}O_{12}$ solid-state electrolyte via microscopy measurements

Can Yildirim [1,7], Florian Flatscher[2,3,7], Steffen Ganschow [4], Alice Lassnig[5], Christoph Gammer [5], Juraj Todt[5,6], Jozef Keckes [5,6] & Daniel Rettenwander [2,3] ✉

Lithium dendrite growth in inorganic solid-state electrolytes acts as a main stumbling block for the commercial development of all-solid-state lithium batteries. Indeed, Li dendrites often lead to solid-state electrolyte fractures, undermining device integrity and safety. Despite the significance of these issues, the mechanisms driving the solid-state electrolyte fracture process at the microscopic level remain poorly understood. Here, via operando optical and ex situ dark field X-ray microscopy measurements of LiSn|single-crystal $Li_{6.5}La_3Zr_{1.5}Ta_{0.5}O_{12}$|LiSn symmetric cells, we provide insights into solid-state electrolyte strain patterns and lattice orientation changes associated with dendrite growth. We report the observation of dislocations in the immediate vicinity of dendrite tips, including one instance where a dislocation is anchored directly to a tip. This latter occurrence in single-crystalline ceramics suggests an interplay between dendrite proliferation and dislocation formation. We speculate that the mechanical stress induced by dendrite expansion triggers dislocation generation. These dislocations seem to influence the fracture process, potentially affecting the directional growth and branching observed in lithium dendrites.

Dendrites present a significant challenge in the development of high-energy Li metal batteries. To to prevent dendrite formation, a deeper understanding of the causes for their nucleation and propagation is crucial. During electrodeposition, uneven metal deposits can lead to dendritic structures with higher local electric fields. This creates a cycle where metal preferentially accumulates in the regions with higher electric fields, causing further dendrite growth[1,2]. In lithium metal batteries using non-aqueous liquid electrolyte solutions, similar challenges arise. To mitigate dendrite growth, extensive research has focused on the utilization of additives and diverse charging protocols, aiming to achieve a more homogeneous lithium deposition[2–6].

Inorganic solid-state electrolytes, on the other hand, were postulated to prevent dendrite growth by creating a mechanical barrier, where the growing soft lithium metal would be halted[7]. Their non-flammability would also reduce safety issues[8]. However, studies have shown that soft lithium metal can still penetrate solid electrolytes, including sulfur-based materials such as $Li_6PS_5Cl$ and mechanically robust oxide-based materials like the garnet $Li_7La_3Zr_2O_{12}$[9–11]. Given the theoretically strong mechanical properties of $Li_7La_3Zr_2O_{12}$ (LLZO), we focus our attention on this material[7,12].

There are multiple explanations reported why dendrites can still penetrate LLZO. One report emphasizes locally enhanced electronic

[1]European Synchrotron Radiation Facility, Grenoble Cedex 9, France. [2]Department of Material Science and Engineering, NTNU Norwegian University of Science and Technology, Trondheim, Norway. [3]Christian Doppler Laboratory for Solid-State Batteries, NTNU Norwegian University of Science and Technology, Trondheim, Norway. [4]Leibniz-Institut für Kristallzüchtung, Berlin, Germany. [5]Austrian Academy of Sciences, Erich Schmid Institute of Materials Science, Leoben, Austria. [6]Chair of Materials Physics, Montanuniversität Leoben, Leoben, Austria. [7]These authors contributed equally: Can Yildirim, Florian Flatscher. ✉e-mail: daniel.rettenwander@ntnu.no

conductivity of LLZO, stemming from chemical inhomogeneities or dopants, which enables the formation of lithium metal by combining an electron and Li⁺ inside the LLZO[13]. Others approach the issue from a mechanical perspective, where preexisting cracks or defects are filled with Li. Insufficient Li flux away from the filled crack during cycling leads to a buildup of hydrostatic pressure which then exceeds the fracture toughness of the ceramic and cracks it open[11,14]. In order for this model to work, the crack needs to be completely filled to allow for the necessary pressure buildup. The exact mechanism of the fracture is still not fully answered and there are open questions such as whether the electrolyte cracks open first, creating more space for lithium metal to fill and allowing the dendrite to continue, or if the growing dendrite ruptures the electrolyte as it progresses[11,15–17]. One argument for the latter is that the yield strength of lithium metal on the nanoscale may be much higher than for bulk lithium[14,18]. The local mechanical integrity of the electrolyte also determines when it fails. Mechanical weaknesses can occur due to chemical inhomogeneities, grain boundaries, surface defects or voids inside the polycrystal[11,19–22]. Grain boundaries, in particular, can exhibit distinct mechanical and conductive properties compared to the bulk material, as it was already calculated for LLZO[23]. Their distribution throughout the solid electrolyte can contribute to the branching of growing cracks, as observed in transparent LLZO where dendrites form in a treelike fashion, reflecting their dendritic origins[22,24]. Grain boundaries can also be a focal point for dislocations[25]. Dislocations in ceramics can be a point of weakness, where it is more likely to fail, or it can be a roadblock in the way of a fracture, where multiple dislocations stop a growing crack in its tracks[21,26–28].

Thus far, experimental investigations of dislocations in solid electrolytes have been limited. This is partly due to the challenges associated with introducing dislocations into ceramics. As they are often sintered the long time at elevated temperatures allows for diffusion to remove dislocations resulting in rather low dislocation densities. To increase the amount of dislocations in sintered ceramics high-temperature plastic deformation is often required. It can also be achieved through a polishing procedure depending on the specific material[28,29]. Another challenge is the limited number of characterization techniques that can resolve and visualize dislocations in ceramic materials. Available methods include Transmission Electron Microscopy (TEM), where the quality of the results can highly depend on the sensitivity of the material to the process, with LLZO being particularly beam sensitive requiring cryo techniques, or X-ray techniques with nano resolution such as Dark Field X-Ray Microscopy (DFXM), a diffraction-based imaging technique[30–35]. In contrast to the limitations of TEM, where the use of ≈100 nm thin foils as samples is required,

DFXM presents an enticing alternative, as it enables the non-destructive characterization of bulk materials, unlocking the ability to explore the intricate 3D strain and orientation states with high detail and precision thanks to a better angular resolution within much larger fields of view (on the order of hundreds of micrometers) compared to that of TEM. In this study, we investigate the immediate environment adjacent to a dendrite formed in single-crystalline $Li_{6.5}La_3Zr_{1.5}Ta_{0.5}O_{12}$, LLZTO, employing operando optical microscopy to monitor dendrite growth and to determine the region of interest, which is then investigated using a combination of DFXM and Focused-Ion Beam Scanning Electron Microscopy (FIB SEM) and TEM. Our results reveal the first-ever observation of a dislocation directly preceding a dendrite in bulk LLZTO.

## Results

Figure 1 schematically shows the synchrotron X-ray measurements. In this work two different modalities of diffraction imaging were used: (i) section DFXM where an X-ray objective focuses and magnifies the diffracted signal onto a Charged Coupled Device (CCD) positioned at around 5 meters away from the sample, as shown in Fig. 1a. For this experiment the (532) reflection was probed and the local lattice orientation and strain around this diffraction vector were scanned using a combined motion of the two sample tilts $\phi$, $\chi$ and $2\theta$ to get a 3D volume as shown in Fig. 1b (ii) nearfield topography with a box beam of 1.2 mm x 1.2 mm (hor × ver) illuminating the sample. The diffracted beam is recorded with a CCD positioned at 35 mm downstream of the sample, illustrated in Fig. 1c.

Figure 2a shows the optical microscopy image of the part of the sample that was used for the synchrotron experiments. The dendrite is observed in darker color in the bright field optical microscopy image. Figure 2b–d shows the near-field topography results from a rocking curve measurement over $\phi$ rotation. Figure 2b shows a background-noise corrected image along a given point of the rocking curve. The edges of the sample along with the shape of the dendrite corresponds well to Fig. 2a. We present a detailed video in Supplementary Materials (Movie 1) showing each acquired raw image at the rocking curve. Figure 2c shows the rocking curve center-of-mass (COM) map showing the local orientation variations of the sample. The overall angular spread of the sample is less then 0.15°, including the contribution from the sample edges that are less homogeneous compared to the interior. One thing of note is that the LLZTO near the dendrite has a certain local orientation (appears in blue) and that the surrounding orientation is different (more towards red). This indicates the local lattice curvature and distortion of the dendrite differs from the surrounding matrix.

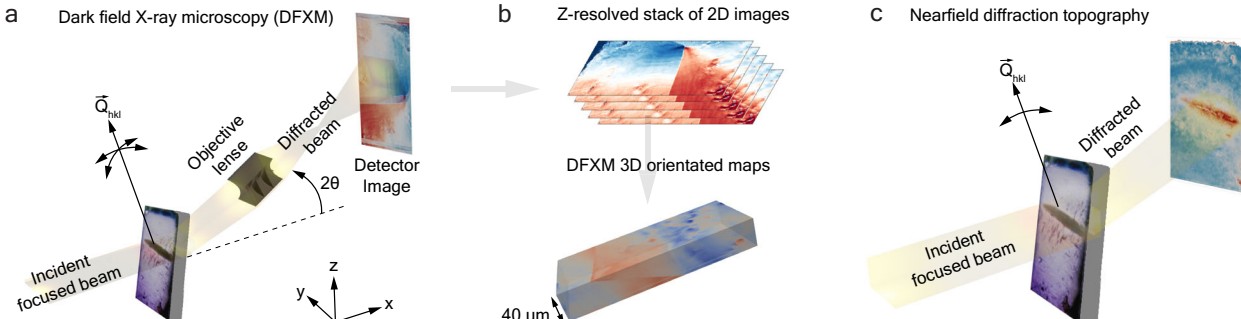

**Fig. 1 | Schematics of diffraction imaging measurements. a** shows the DFXM setup. This study focuses on the observation of dislocation structures at the tip of a dendrite projected from the 532 Bragg reflection using diffraction contrast to obtain comprehensive 3D information. **b** DFXM images were collected for 2D layers by employing a line-focused beam with a beam profile of 600 nm full width at half maximum (FWHM) in the $z$ direction. The sample was scanned vertically in the $z$ direction to capture variations along the crystal's height. The direction of the

scattering vector can be adjusted through the two tilts ($\phi$ and $\chi$), and the length of the scattering vector can be modified using a combined $\phi$-$2\theta$ scan. **c** shows the schematics of nearfield topography measurements with a box-shaped incident beam. This imaging modality provides a larger field of view (1.2 mm × 1.2 mm) with lower spatial resolution (on the order of 1.5 μm) and no $z$ resolution, as the entire diffracted volume is projected onto the nearfield detector.

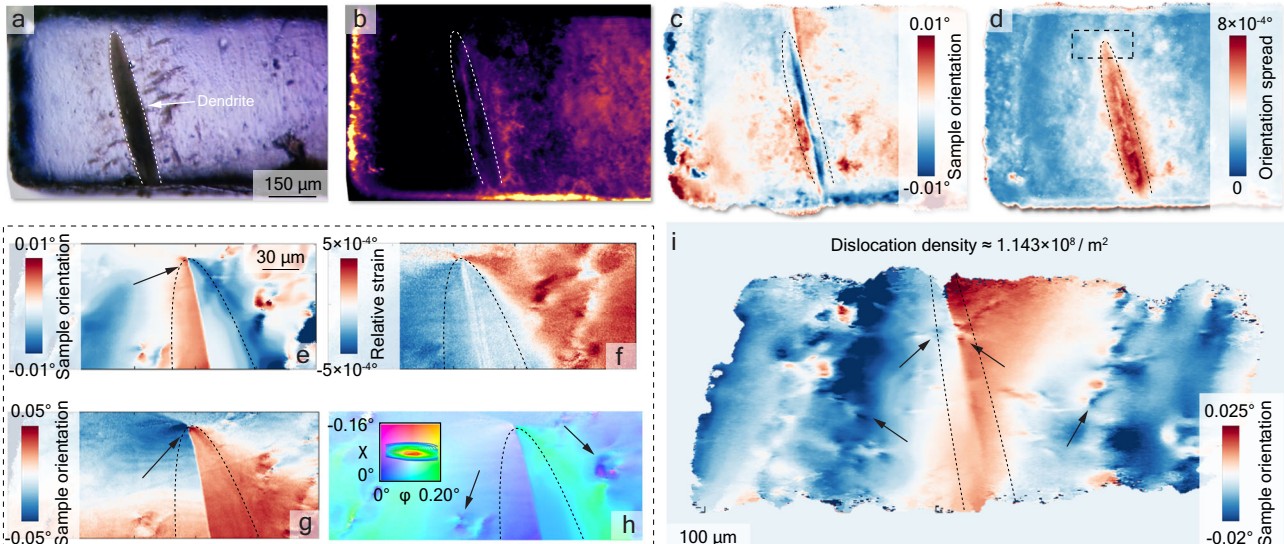

**Fig. 2 | Comprehensive Characterization of Dislocations around Dendrite in LLZTO Crystal.** Large field of view characterization of the LLZTO crystal containing the dendrite (**a**) optical microscope image of the LLZTO, the dendrite being the black shape almost completely passing through the solid electrolyte. **b**–**d** show the nearfield projection maps from a rocking curve with box beam illumination. **b** A background corrected raw image of the crystal in the nearfield projection (**c**) generated center of mass map of the rocking curve scan indicating local orientation distribution (**d**) generated full width half maximum (FWHM) map of the rocking curve showing that the dendrite has a high angular spread. The dashed rectangle marks the region of interest to which we zoomed in on using DFXM, shown in **e**–**h**. DFXM maps zooming in on the the dendrite tip, shown with the dashed line, using a line-focused beam. These maps show 2D sections of the volume from orientation (mosaicity) and strain scans. **e** COM map of the rocking tiltb (**f**) strain map (**g**) COM map of the rolling tilt (**h**) mosaicity map of the two sample tilts and the color key shows the intensity distribution of the 2D mosaicity scan overlaid. **i** COM map of the rocking tilt over a large field of view using the 2× objective on the far-field camera, illustrating the distribution of dislocations, with some marked by black arrows. The area of the dendrite tip in (**e**–**h**) and the dendrite in (**i**) is highlighted by dotted lines.

Figure 2d shows the generated full width at half maximum (FWHM) map of the rocking curve. The orientation spread around the dendrite is significantly higher than the matrix itself. Moreover, the effect of the inhomogeneous distribution of orientation is spread over more than 350 μm. These near-field topography maps with the box beam (rocking curve projection) provides a useful first-glance characterization and overview for the sample and the dendrite. Projection maps, however, integrate the diffracted signal across the entire thickness of the sample, resulting in a lack of height information in the $z$ direction. To overcome this limitation, we utilize DFXM section topography measurements. In this approach, the sample is scanned in the $z$ direction while being illuminated by a line-focused beam. By stacking these 2D virtual slices, we can construct a comprehensive 3D map of the crystal[36].

Figure 2e–h shows the section DFXM maps of sample mosaicity and strain. These maps show the same 2D layer with 600 nm thickness in the $z$ direction within the bulk. A 2D section of the dendrite can be seen as a line in Fig. 2e, which reveals the intricate orientation distribution over $\phi$ rotation with a spatial resolution of 100 nm. Similar to the nearfield projection results, the DFXM results show that there is a certain orientation inhomogeneity around the dendrite. In fact, dendrite acts as a boundary between positive and negative orientation distribution with respect to the peak of the rocking curve. Moreover, there are isolated individual dislocations within the field of view. A noteworthy observation is marked by the black arrow: a dislocation that pins the edge of the dendrite. Similarly, in the $\chi$ COM map, Fig. 2g, the dendrite acts as a boundary that separates zones with different local orientation distribution. The dislocation that is at the edge of the dendrite is more prominent in the COM map, manifested by positive and negative orientation around the dislocation. The strain map in Fig. 2f shows that there are compressive and tensile strain regions at each side of the dendrite. This strain polarization is marked at the edge of the dendrite where the dislocation is located. The measured elastic strain values reach as high as 0.05%, corresponding to stress values of $\approx$ 77 MPa (Elastic constant of LLZO at 298 K 154.5 GPa)[7].

## Discussion

The typically low dislocation densities in single-crystalline ceramics, as minimal as $10^6\,\mathrm{m}^{-2}$, arise from uniform temperature control and extended periods at high temperatures that facilitate diffusion during processing[37]. The calculated dislocation density around the dendrite tip, shown in Fig. 2e–h is approximately $4.42 \times 10^8\,\mathrm{m}^{-2}$ determined from ~ 12 dislocations in the $102 \times 246\,\mu\mathrm{m}^2$ area. Figure 2(i), which shows a lower magnification of the dendrite demonstrates a lower dislocation density of approximately $1.14 \times 10^8\,\mathrm{m}^{-2}$, with ~ 20 dislocations populating a $250 \times 700\,\mu\mathrm{m}^2$ area. Some of them are highlighted with black arrows in Fig. 2i. It has to be noted that these calculations represent an underestimation for the dislocation density as there may be more dislocations in this region that could not be detected from the probed reflections. The regions farther from the dendrite, where dislocations are sparse, can be considered closer to a pristine single crystal. TEM characterization (Supplementary Fig. 1) showed no observation of dislocations, though it is likely that a pristine region was observed as the reduced visibility of the dendrite in the electron microscope compared to the optical image causes accurate positioning of the FIB lamella to be a non trivial problem. Given the low dislocation density in these pristine regions, the probability that a dendrite tip would encounter a dislocation precisely at the moment the experiment was stopped is exceedingly low. Therefore, the dislocations seen in Fig. 2e–h near dendrites are not accidental but related to the dendrites, indicating a significant link between processing conditions and dislocation formation[38]. This relationship provides a solid foundation for understanding the material's structural characteristics. Another point that reinforces their connection is their location close to the dendrite, with one of them, marked by a black arrow pinning the tip of the dendrite in Fig. 2e. The inlaid box in Fig. 2d shows the area surrounding the dendrite tip seen in the far field image in Fig. 2e–h. The position of the dislocations marked with black arrows inf Fig. 2h appear to be placed in a way that matches the branching pattern of secondary dendrites coming from the main fracture. The correlation between the optical, the near field and the far field image is

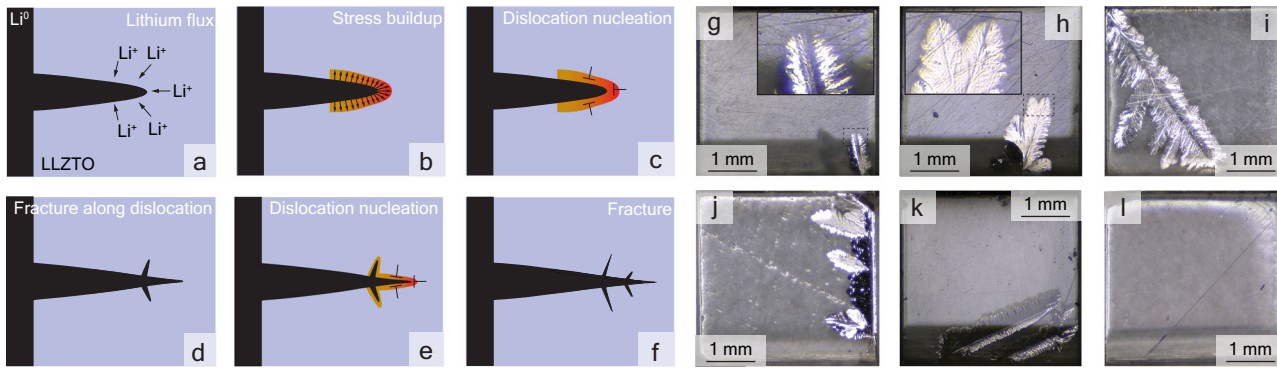

**Fig. 3 | Mechanism of Dislocation Nucleation and Fracture in LLZTO with Observed Dendrite Growth Patterns (a–f) Proposed mechanism with schematics for the dislocation nucleation and fracture in LLZTO. a** High current is applied causing increased Li⁺ flux towards a lithium filled flaw, based on Porz et al.[11] **b** The filled flaw exerts pressure on the LLZTO and causes the stress (zone drawn in orange) to increase. **c** The high stress causes dislocations to nucleate. Note that while the symbol for edge dislocations is used for illustration the exact nature is unkown. **d** Fracturing of the LLZTO and the crack fills with lithium. Note here that the precise order and timing of the events is not known. The stress zone adapts to the new cracks (**e**) new dislocations nucleate in the stressed zone. **f** Cracks grow further until the current is switched off, extinguishing almost all dislocations. **g–l** Optical microscopy images of dendrites created in single crystalline LLZTO, not investigated with DFXM, using a current density of 4.5–5 mA/cm². The growth direction is from bottom to top with the dendrite growing along the edge in (**j**) and branching into the center. A leaflike branching structure occurs in (**g**, **k**). In (**l**) no branching occurs. It is notable that the dendrites do not seem to take the shortest path to the other electrode, which are at the top and bottom of the electrolyte. The setup used is shown in Supplementary Fig. 3.

necessary as the layer view in the far field image only captures a part of the 3D growth of the dendrite.

In contrast to metals, the mobility of dislocations in ceramics at 298K is rather low, often in the nm/s or $\mu$m/s range. Moreover, increasing stress levels frequently do not result in accelerated dislocation movement; instead, they tend to induce fractures within the ceramic material[37]. It is therefore unlikely that the dislocation is able to outpace the growing dendrite, which at the current densities of a few mA/cm² employed here can be in the range of 50 $\mu$m/s[39,40]. We hypothesize that the stress originating from dendrite growth initiates the formation of dislocations, and the persistent dislocations are a result of the cessation of dendrite growth. The process is described in Fig. 3a–f.

The stress needed to nucleate a dislocation in ceramics can be in the GPa range, as reported for SrTiO₃[41]. Deformation experiments in silicate garnets requires a pressure of 6.5 GPa at 700 °C to introduce dislocations without fracture. However, more recent experiments using single crystalline LLZTO achieved plastic deformation at 1000 °C with just 100 MPa of pressure. Nonetheless, uncertainty remains about whether this outcome arises from dislocations or another competing mechanism[28,42]. At room temperature the stress needed to mobilize dislocation is likely to be higher. The stress at the dendrite tip during fracture can reach GPa levels according to Gao et al. while the stress surrounding the dendrite tip during growth is shown on the order of -750 MPa[40,43]. It seems therefore feasible for the local stress to reach levels where a dislocation nucleates. While in our DFXM experiments, the remaining stress in the material was measured to be approximately 77 MPa. This indicates that the material was able to relax between the growth of the dendrite and the DFXM measurement. The measured strain values at different heights of the sample using the line-focused beam support these findings, as shown in Supplementary Fig. 2. Additional operando measurements where we have a thorough characterization of the initial state before the dendrite growth and could follow dendrite evolution operando using DFXM could help prove the actual stress nucleating the dislocation at the dendrite tip and determine whether the dendrite moves along the dislocation path or not. Note that one would need multiple Bragg reflections to fully assess the complete stress state. In our measurements, we measure the strain state around a given diffraction vector.

Our hypothesis posits that dislocations serve as preferential sites for vulnerability within the material, thereby creating energetically favorable pathways for fracture propagation. This theory could elucidate the presence of elongated, leaf-like structures observed in Fig. 2a. Such morphologies have been consistently replicated in single crystalline LLZTO under high current densities (1–10 mA/cm²), as depicted in Fig. 3g–l, where the dendrite grew from the bottom to the top of the electrolyte. Notably, in some instances, such as shown in Fig. 3l, dendritic branching does not occur, suggesting a complex interplay between dislocation distribution and dendritic growth mechanisms.

The dendritic morphology manifested within the material may plausibly arise from the heterogeneous distribution of electrical current, consequently resulting in preferential lithium deposition at sites of elevated local current density. Theoretical work for liquid electrolytes shows that an increase in current density transitions a singular growth to a branching structure[44]. Phase field simulations for Li deposition in liquid as well as solid electrolytes also demonstrate that branching structures are possible[45,46]. However, in the referenced simulations, it is assumed that dendritic growth occurs wherever Li is deposited, naturally favoring the shortest route to the opposing electrode due to enhanced current flow resulting from decreased separation. Contrary to this assumption, Fig. 3i, k, l demonstrates dendritic growth occurring at an angle, diverging from the expected direct path predicted by current distribution models. This discrepancy suggests the necessity of incorporating a mechanical perspective, specifically how dendrites navigate through the solid electrolyte by creating fractures to facilitate their expansion.

Building on the framework proposed by Porz et al., the mechanism of fracture is conceptualized as being driven by hydrostatic pressure at the lithium-saturated crack tip, where the influx of Li surpasses its efflux, generating sufficient force to breach the electrolyte[11]. This model elucidates the intertwined roles of electrochemical and mechanical forces in dendrite growth, emphasizing their intricate interplay within the solid electrolyte environment. Here, the applied current modulates lithium flux, directly affecting the buildup of pressure. Fracture ensues when the stress surpasses the material's fracture toughness, $K_{IC}$, emanating radially from the crack's origin. The phenomenon of fractures propagating in multiple directions simultaneously, thereby dispersing stress throughout the material, remains an area of inquiry. This stress dispersion potentially hinders further fracturing unless countered by a significant, localized force. However, in scenarios where significant stress is present, dislocations are likely to form with restricted mobility due to the scarcity of slip planes in

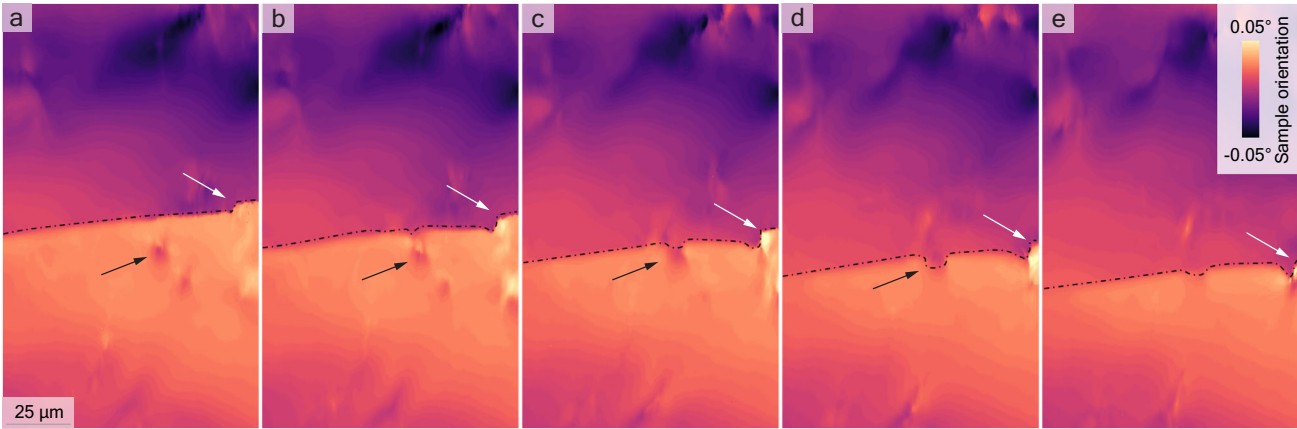

**Fig. 4 | 3D High-Resolution DFXM Orientation Maps Revealing Dendrite-Dislocation Interactions in LLZTO.** DFXM orientation maps (**a**−**e**) with 2 μm spacing in *z* direction, highlighting dendrite-dislocation interactions. The dashed line outlines the dendrite plane, while arrows indicate dislocations altering the dendrite shape within the material's bulk.

ceramic materials[47]. Should a crack propagate along these dislocations, it enables fractures to branch out in multiple directions simultaneously, creating a pattern of repetition along the main crack. This behavior underscores the dynamic interplay between dislocations and crack propagation, offering valuable perspectives on the underlying mechanics governing fracture patterns in ceramics. Figure 4 displays a series of 2D DFXM orientation maps with a 2 μm inter-slice spacing, revealing the interactions between dendrites and dislocations within a material. In Fig. 4a and b we can see dislocations below the dendrites marked by the black arrow. On Fig. 4c, d, and e, these dislocations interact with the dendrite as marked by the dashed lines, which trace the dendrite planes, while arrows point to dislocations that modify the dendrite shape in the bulk. A full video (Supplementary Movie 2) covering a 40 μm distance in *z* is available in the Supplementary Material. Looking at this with the goal of minimizing the occurrence of dendrites brings up a few challenges and possible pathways for material optimization. The observation that growing dendrites can lead to dislocations, potentially weakening the material, underscores a significant issue. Understanding the precise conditions that cause these dislocations requires detailed, real-time measurements. This approach is crucial for maintaining the material's strength while addressing the challenges posed by dendrite propagation. Especially as there are few ways to control the local current distribution inside of the electrolyte to influence the stress near the fracture. One available option is to reduce the current densities used to keep local currents and therefore the stress at low levels though this contradicts the goal to achieve fast charging with solid electrolytes. Other avenues are to increase the fracture toughness of the material, either by introducing residual stresses or, seemingly counter intuitive, by increasing the amount of dislocations in the material[27,48]. A higher dislocation density can allow for plastic deformation of the material via dislocation movement as a competing mechanism to fracture, raising its fracture toughness. For ceramic electrolytes this can be achieved in multiple ways. Shortening the annealing period during the sintering process is one approach. Typically, prolonged exposure to high temperatures during sintering facilitates atom diffusion and reduces dislocations. However, employing rapid sintering methods like flash sintering or blacklight sintering can significantly speed up the process, reducing it to mere minutes or seconds. This limited time frame restricts diffusion, helping to retain dislocations that can enhance the material's properties[49–51]. The other options is to increase the entropy in ceramics, which has already been shown to increase the dislocation density as the strain energy caused by dislocations is compensated by the configuration entropy[52]. There are already examples of the synthesis of high entropy LLZO and other high entropy solid electrolytes with the

high entropy LLZO showing a long term cycling stability[53,54]. However, to conclusively determine whether the observed improvements are due to increased dislocation density, further experimental research on high entropy materials is essential.

In summary, in this study, we report the observation of dislocations at the tip of growing dendrites within bulk LLZTO samples. By utilizing Dark Field X-ray Microscopy, we have closely examined the micro-environment around the dendrite tips, shedding light on their growth behavior. This finding suggests an intriguing link between dendrite growth and dislocation formation. Notably, the characteristics of low dislocation mobility and density in ceramics led us to infer that these dislocations could originate from the stress induced by dendrite expansion. Due to the low dislocation mobility and low dislocation density in ceramics we conclude that the dislocations are likely connected to the dendrite and nucleate from the stress emitted by its growth. Those dislocations could also pose a weakness in the material allowing for easier fracture in certain directions, allowing for the dendrite to branch out. Looking ahead, carrying out additional operando DFXM measurements could offer deeper insights into the dynamics of dendrite growth and deformation, helping to validate the proposed connection between dendrites and dislocations. Additionally, the fundamental investigations presented in this article could benefit future research work on manipulating dislocation density in ceramics to potentially curb dendrite growth Strategies like enhancing material entropy or employing advanced sintering methods might provide avenues to explore this avenue further.

## Methods
### Samples
Czochralski drawn $Li_{6.5}La_3Zr_{1.5}Ta_{0.5}O_{12}$ single crystals were oriented with Laue Diffraction and cut with a diamond wire saw into $3 \times 2 \times 0.5$ mm cuboids. The $3 \times 2$ mm sides were mechanically thinned to 200 μm thickness and polished with SiC grinding paper with P1200, P2400 and P4000 grit size and final polishing using diamond paste of 3, 1 and 0.25 μm particle size. The samples were put in a 0.1M HCl solution for around 20 s to remove $Li_2CO_3$ contamination layers, rinsed with isopropyl alcohol and transferred into an Ar filled glovebox ($O_2$ and $H_2O$ levels below <1 ppm). Two opposite 200 μm × 3 mm sides were coated with a molten LiSn (30 wt% Sn) alloy. The coated sample was placed in a homemade setup, where it was put on between two brass current collectors, with a white foam below for contrast, which were closed via a screw. The estimated pressure in the setup is below 5.5 kPa, which is three orders of magnitude below what is used during battery assembly[55]. The measurement setup is shown in Supplementary Fig. 3.

## Operando optical microscopy measurements

After focusing on the region of interest with the optical microscope a current density between 1 and 10 mA/cm$^2$ was applied for 5–20 s at 25 ± 1 °C to the symmetric LiSn|LLZTO|LiSn cell using a Solartron Modulab XM ECS, to facilitate dendrite propagation. The progress of the dendrite in the solid electrolyte was observed via optical microscopy and the process was being captured on video. The current was stopped when the dendrite reached around halfway through the electrolyte. After the dendrite was grown the samples were placed in a membrane box and vacuum sealed for shipping to the ESRF. There the samples were stored in an Argon filled glovebox $O_2$ and $H_2O$ levels below <0.5 ppm) until the measurement.

## Ex situ dark field X-ray microscopy measurements

DFXM experiments were performed at the ID06-HXM beamline of the European Synchrotron Radiation Facility[33]. Shortly before the measurement the LLZTO containing dendrites was removed from the Ar-filled glovebox and placed on the mounting bracket. During the whole measurement (8 h) the sample is exposed to air, though the effect on the region of interest, the dendrites and surrounding region within the LLZTO, should be negligible. Using 17 keV photons selected by a Si monochromator with a ΔE/E bandwidth of $10^{-4}$, the beam was focused vertically using a Compound Refractive Lens (CRL) comprising 58 1D Be lenslets. The effective focal length was 72 cm, yielding a horizontal and vertical beam profile of approximately $200 \times 0.6$ μm$^2$ (FWHM), respectively. A single plane through the crystal was illuminated by the horizontal line beam, defining the observation plane of the microscope. After near-field alignment, the near-field camera was removed, and the image was magnified by an X-ray objective lens consisting of 88 Be parabolic lenslets. The CRL was aligned 269 mm from the sample along the diffracted beam using a far-field detector. With an X-ray magnification of Mx = 17.9× , the far-field detector employed an indirect X-ray detection scheme and a visible microscope with a 2160 × 2560 pixel PCO.edge sCMOS camera. Positioned 5030 mm from the sample, the visible optics in the far-field detector provided two objective options (10× and 2× ) for an effective pixel size of 0.75 μm or 3.75 μm, respectively. This study focuses on the analysis of 10× magnification images (M = 179×), therefore most of the far-field DFXM images shown in this manuscript are at 10× magnification unless otherwise mentioned. Three scan types were conducted: rocking scans, mosaicity scans, and axial strain scans. Rocking scans involved a tilt angle $\phi$ range of $\Delta\phi = 0.15°$ in 25 steps (0.006° per step) to map displacement gradient tensor field components. Mosaicity scans and axial strain scans assessed distortions along orthogonal tilts $\chi$ and $\phi$, and the $2\theta$ axis, respectively. Voxels were associated with (HKL) pole figure subsets, allowing the generation of Center of Mass (COM) maps for voxel-level (HKL) orientation. Finally, axial strain scans quantified residual strain by scanning the $2\theta$ axis ($\Delta 2\theta = 0.01°$) and were reconstructed into the COM maps.

## Data availability

The raw data generated in this study have been deposited in the ESRF database under accession code https://doi.org/10.15151/ESRF-ES-992881230and are available under restricted access due to data privacy laws. Access can be obtained upon request. The processed data are available under restricted access at this link and can be provided upon request.

## Code availability

The analysis tools used for data evaluation are available at GitLab ESRF and GitHub.

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

## Acknowledgements
We thank Lukas Porz for his valuable contribution to the design of the experiment and helpful discussions. Marta Mirolo and Valentin Vinci are acknowledged for helpful discussions and proof read. The authors are grateful for provision of beamtime at ID06-HXM. D.R. and F.F. acknowledge financial support by the Austrian Federal Ministry for Digital and Economic Affairs, the National Foundation for Research, Technology and Development and the Christian Doppler Research Association (Christian Doppler Laboratory for Solid State Batteries). C.Y. acknowledges the financial support by the ERC Starting Grant "D-REX" (no 101116911). D.R. and J.K. acknowledge funding under the scope of the COMET program within the K2 Center "Integrated Computational Material, Process and Product Engineering (IC-MPPE)" (Project ASSESS P1.10) and from the European Union's Horizon Europe research and innovation program under Grant Agreement No 101103834 (OPERA).

## Author contributions
D.R. conceived and designed the project with help from J.K. F.F. performed the experimental work. S.G. synthesized the single crystals. A.L. performed the TEM measurements with help from C.G. C.Y. performed the DFXM measurements and data analysis. F.F., C.Y., and D.R. wrote the first draft with discussion and feedback from J.T. and J.K. All authors contributed to the final draft.

## Funding
 Olavs Hospital - Trondheim University Hospital).

## Competing interests
The authors declare no competing interests.
