## [Peer Review File · Nature Communications]

REVIEWER COMMENTS

Reviewer #1 (Remarks to the Author):

In this manuscript, the author uses a Dark Field X-ray Microscopy to provide high-resolution insights into strain patterns and lattice orientation changes associated with dendrite growth in LLZTO solid-state battery. The author associates the mechanical stress, dislocations and dendrite growth. This is an interesting phenomenon. There are several points, which need further attention and significant revision.

1. The author may have used a model battery, which can observe the growth of lithium dendrite through optical windows. However, the author does not show the structure of the model battery.
2. In Figure 1 or Figure 3, the author only gives pictures that seems to be lithium dendrite piercing LLZTO, but the field of view is too small and does not understand the electrode structure.
3. In Figure 1, is the result of the test disassembled battery testing or non-destructive testing? If it is the test of disassembling the battery, how to ensure that the sample transfer process will not affect the lithium dendrites. Lithium dendrites are known to be very sensitive to the environment. In short, the authors reveal too few details of the experiment.
4. In addition, the length of lithium dendrite in Figure 1 are about 300 μm , while the lithium dendrite in Figure 3 g-k are 1000-5000 μm . Please confirm whether the batteries of the same system are used.
5. For comparison, the author should give the relevant characterization of the original battery not cycling.
6. The author claims that the low dislocation densities of the single crystal ceramics is as low as 10^{-6} m^{-2} , the dislocations seen in Fig. 2(e-h) near dendrites are not accidental but related to the dendrites. However, the manuscript did not give the relevant detection of LLZTO's dislocation densities. This is important to conclusions in this manuscript.

Reviewer #2 (Remarks to the Author):

The authors used a technique called Dark Field X-ray Microscopy (DFXM) to identify the defect structure near lithium dendrites in LLZTO. Although it is somewhat new to use DFXM to study LLZTO, the result obtained from such technique is limited. There are several deficiencies in this manuscript.

1. It is questionable whether the observed contrast is from a dislocation. The strain contrast from a dislocation falls in nanometer range, which may be beyond the resolution of DFXM which is about 100 nm as the authors claimed. In fact, the prevalent defects during the dendrite propagation are cracks, not dislocations. I am surprised that no cracks were observed. Given the optical image showing the size of the dendrite is huge, it is unlikely that there are no cracks.
2. Even if the observed defect is dislocation, its role on dendrite growth may not be significant, neither to the electrochemical performance. Therefore its impact to the battery community is limited.
3. There is lack of experimental details, e.g. what was the experimental set-up for the dendrite growth? How was the electrochemical experiment was done? What sample was used for the study?

Reviewer #3 (Remarks to the Author):

The article titled "Origin of dendrite branching in solid-state batteries" by Yildirim and Flatscher et al. present findings on the interplay between the formation of dislocation and propagation of lithium dendrite in single crystal garnet-type LLZTO solid electrolyte using Dark-field X-ray Microscopy (DFXM). According to the presented results, dislocations are formed within the solid electrolyte due to the build-up of hydrostatic pressure caused by electric-field driven deposition of lithium at a dendrite. The dislocations formed in the material create a weakness, leading to preferential cracking of the LLZTO and resulting in branching of the dendrite structure.

The manuscript raises interesting claims and discussions about the mechanistic understanding of dendrite propagation within single crystal LLZTO, which are highly valuable for the solid-state battery community. Additionally, the visualization of dislocations and their correlation with dendritic structures is a significant breakthrough going beyond the current state of research. However, the conclusions drawn base only on a single post-mortem experiment, leaving much space for miss interpretation. As no initial DFXM was captured prior dendrite formation, it remains ambiguous whether dendrites propagation initiates dislocation formation or dendrites propagate already existing dislocation within the LLZTO.

Therefore, the manuscript is recommended for publication after addressing the mentioned concern. Additionally the following points require further explanation or correction:

- a. The actual title is misleading. The manuscript focuses within the introduction as well as the rest of the paper on Li batteries. I.e., no literature about solid electrolytes for Na batteries is taken into account. Furthermore no real battery was assembled for the electrochemical experiments (only half cell). So, it might be fair to adapt the title to “Origin of dendrite branching in Li solid-state battery materials”. – Or alternatively expand to other types of solid state batteries.
- b. While DFXM have been used in material science the method is rather new in the field of solid-state batteries. It is necessary to explain how dislocation can be visualized by this method. Moreover, are the observed dislocations single dislocations or an average of several dislocations within a small voxel?
- c. Why does the sample orientation in Figure 2e vary significantly along the highlighted dendrite starting from the tip inside the “bulk” of the Li?
- d. Why is the sample orientation in Figure 2g opposite to the sample orientation given in Figure 2e?
- e. Is it possible to further categorize the observed dislocation as screw or edge dislocation? If so, what type have been observed at the tip of the dendrite as well as on the branching parts?
- f. Did you also observe dislocations in the sample investigated which are not located near the dendrite to support the statement of low dislocation density?
- g. Since the manuscript is based on one post-mortem analysis only, a reference sample without dendrites should be measured in comparison and data given in the SI. If there are dendrite free areas within the investigated sample, these can also be taken as a reference.
- h. Why does dendrite growth go from right to left in Figure 3j, when the electrode is placed on top and bottom?
- i. The authors suggest to increase the fracture toughness by increasing the dislocation density of the LLZTO as dislocation movement allows plastic deformation and thus pressure release. However, as outlined in the discussion the dislocation movement in ceramic is comparably low. So, would this be a good solution with respect to the kinetic time scale?
- j. Statement of the “perfect single crystal” in the caption of Figure 1 of the SI should be reevaluated, as TEM only gives a local point of view.

Additionally smaller comments:

- a. Please check the chronological order in the first paragraph of the introduction. Maybe start to address liquid electrolytes and move to solid electrolytes. - The phrase “Similar challenges” does not fit in the sense of the previous and following sentence.
- b. Please refer in the main text to the figures shown in the Supplementary Information

c. The term “dendrite formation” is too general as it include the initial nucleation as well as the propagation process. It ´s suggested to use more specific terms to describe the observed phenomena, like “dendrite initiation” and “dendrite propagation”.

d. Please highlight the dendrite structure in the Figure 2b, c and d to help the reader identifying the mentioned “certain local orientation of LLZTO” and the surrounding.

e. We observed issues with opening supplementary video 2, as it is coded with png. data type.

f. P. 5: “in-operando measurement” is not existing. In contrast to in-situ the right wording is “operando measurement”.

T. Ortmann and M. Rohnke

Reviewer #4 (Remarks to the Author):

I co-reviewed this manuscript with one of the reviewers who provided the listed reports. This is part of the Nature Communications initiative to facilitate training in peer review and to provide appropriate recognition for Early Career Researchers who co-review manuscripts

RESPONSE LETTER AND THE LIST OF CHANGES IN THE REVISE MANUSCRIPT NCOMMS-24-10130

Reviewer #1:

In this manuscript, the author uses a Dark Field X-ray Microscopy to provide high-resolution insights into strain patterns and lattice orientation changes associated with dendrite growth in LLZTO solid-state battery. The author associates the mechanical stress, dislocations and dendrite growth. This is an interesting phenomenon. There are several points, which need further attention and significant revision.

1. The author may have used a model battery, which can observe the growth of lithium dendrite through optical windows. However, the author does not show the structure of the model battery.

Thank you for bringing this up. The structure of the cell is mentioned in the methods section as a symmetric Li|LLZTO|Li cell. What appears to be an optical window is actually the solid electrolyte. We have updated the methods section to clarify this and provide a clearer explanation for the reader.

2. In Figure 1 or Figure 3, the author only gives pictures that seems to be lithium dendrite piercing LLZTO, but the field of view is too small and does not understand the electrode structure.

As mentioned in response to question 1 and detailed in the methods section, the cell structure used in our experiments was a symmetric Li|LLZTO|Li configuration. Figure 1 shows part of the LLZTO with the region of interest that was observed in the DFXM measurement. Figure 3 on the other hand has a top down view on the whole Li|LLZTO|Li configuration which is clamped in the setup which is now added in the supplementary information.

3. In Figure 1, is the result of the test disassembled battery testing or non-destructive testing? If it is the test of disassembling the battery, how to ensure that the sample transfer process will not affect the lithium dendrites. Lithium dendrites are known to be very sensitive to the environment. In short, the authors reveal too few details of the experiment.

Thank you for your comment. Firstly, we would like to highlight that the primary objective of our measurement is to assess the stress within the LLZO, not within the lithium metal. Therefore, any degradation of the lithium dendrites should not significantly affect the measurement results. To minimize potential degradation, the samples were placed in a membrane box and vacuum sealed for transport immediately after dendrite growth. Shortly before the measurement, the samples were removed from the glovebox. Although the exterior of the LLZTO may come into contact with air and slightly degrade, the dendrite is encased in the solid electrolyte and is thus protected from ambient exposure. We have updated the methods section to better reflect these precautions and the non-destructive nature of the testing.

4. In addition, the length of lithium dendrite in Figure 1 are about 300 μm , while the lithium dendrite in Figure 3 g-k are 1000-5000 μm . Please confirm whether the batteries of the same system are used.

The image in Figure 1 is a close-up of a dendrite tip within LLZTO. The dendrites shown in Figures 3g-k are from LLZTO samples, which were not investigated with DFXM but still belong to the same $\text{Li}|\text{LLZTO}|\text{Li}$ system. We have updated the caption of Figure 3 to better reflect this.

5. For comparison, the author should give the relevant characterization of the original battery not cycling.

We acknowledge the importance of comparing the dislocation densities in cycled versus non-cycled batteries. For the pristine, non-cycled battery, the dislocation density should be notably low, estimated to be around 10^6 m^{-2} . As we do not have measurements of the region before the dendrite growth we have included an image with lower 2x magnification in the new Figure 2 (i) depicting the COM map of the dendrite and the further surrounding regions. This figure shows a volume of approximately $700 \times 250 \times 0.6 \text{ \mu m}^3$, indicating that there are fewer dislocations in this region compared to the dendrite tip itself. The measured dislocation density in this new figure is on the order of $\sim 1.14 \times 10^8 \text{ m}^{-2}$, which is around 4 times lower compared to the dendrite tip.

Moreover, the further away from the dendrite, the fewer dislocations are observed, thus these regions can be considered as the pristine parts of the sample. The probability of encountering a dislocation precisely at the dendrite tip during the snapshot is extremely low, reinforcing that dislocations are predominantly located around the dendrite tip rather than farther away. This spatial relationship supports the idea that regions distant from the dendrites represent pristine samples and can be considered as an alternative to different samples.

Additionally, it is important to note that the beamline ID06-HXM was undergoing an upgrade (now operational since a couple of months). We are planning to perform operando experiments to showcase the origin and propagation events of dislocations and dendrites in real-time. This will be the focus of a subsequent study and will provide further insights into the dynamic processes involved. We hope these clarifications and additional data adequately address the concern and provide a clear comparison of the characterization of the original battery not cycling.

6. The author claims that the low dislocation densities of the single crystal ceramics is as low as 10^6 m^{-2} , the dislocations seen in Fig. 2(e-h) near dendrites are not accidental but related to the dendrites. However, the manuscript did not give the relevant detection of LLZTO's dislocation densities. This is important to conclusions in this manuscript.

Thank you for mentioning this. The text has been updated with the dislocation density near the dendrite tip (shown in Fig.2e-h) which is $\sim 4.42 \times 10^8 \text{ m}^{-2}$ determined from ~ 12 dislocations in the area of $102 \times 246 \text{ \mu m}^2$. This is 2 orders of magnitude higher than what we could expect from a pristine sample. We have also updated figure 2 with figure 2i showing a lower 2x magnification

image of the COM map of the rocking curve which also shows the area further away from the dendrite. There the dislocation density was determined as $\sim 1.14 \times 10^8 \text{ m}^{-2}$ with ~ 20 dislocations in an area of $250 \times 700 \text{ }\mu\text{m}^2$. It has to be noted that these calculations represent an underestimation for the dislocation density as there may be more dislocations in this region that could not be detected from the probed reflections. This reduction in the dislocation density when moving away from the dendrite supports the conclusion that the dendrite and the dislocations are indeed connected.

Reviewer #2:

The authors used a technique called Dark Field X-ray Microscopy (DFXM) to identify the defect structure near lithium dendrites in LLZTO. Although it is somewhat new to use DFXM to study LLZTO, the result obtained from such technique is limited. There are several deficiencies in this manuscript.

1. It is questionable whether the observed contrast is from a dislocation. The strain contrast from a dislocation falls in nanometer range, which may be beyond the resolution of DFXM which is about 100 nm as the authors claimed. In fact, the prevalent defects during the dendrite propagation are cracks, not dislocations. I am surprised that no cracks were observed. Given the optical image showing the size of the dendrite is huge, it is unlikely that there are no cracks.

We appreciate the author's comments regarding the observed contrast in our DFXM images and the identification of dislocations, and we find them valuable in improving the manuscript's clarity. Dislocations generate distinctive strain fields that exhibit a $1/r$ decay. The distance over which dislocation strain fields can be observed therefore depends only on the strain resolution, in our case the angular resolution. As shown in Figure R1, in DFXM images dislocation strains are visible as dark-bright contrast transitions due to changes in local lattice plane orientations and spacings. This signature pattern differentiates dislocation strain fields from cracks, which have different contrast patterns due to their distinct strain gradients. The contrast observed in our images at the dendrite tips matches the strain fields from dislocations, differing from cracks. The strain fields around dislocations produce characteristic intensity patterns due to lattice plane distortions, which are detectable in DFXM despite spatial resolution limitations, *thanks to the high angular resolution of the technique*. While it is true that the dislocation core is below the spatial resolution limit, we observe the strain fields extending micrometers, as shown in Figure R1.

Over the years, we have experimentally demonstrated extensive networks of dislocations in various materials, such as semiconductors and ceramics, using DFXM and X-ray topography. Studies by Yildirim et al. (2021, J. Synch Rad; 2023, Sci. Rep.; 2023, J. Mat. Sci) and Simons et al. (2019, ACS Nano Letters) provide empirical evidence of these techniques' effectiveness in revealing dislocation networks despite spatial resolution constraints. Moreover, Borgi et al. (2024, JAC) have elucidated the contrast mechanisms of dislocations in DFXM, showing through simulations how strain fields around dislocations can be effectively captured due to the method's high angular sensitivity. Poulsen et al. (2021, JAC) presented a geometrical optics formalism that offers a theoretical foundation for understanding strain-induced contrast in DFXM. Based on these well-established theoretical and experimental protocols, it is clear that the observed contrast at

the tip of the dendrite (along with some other specific locations around the dendrite shown in Figure 5 in the original manuscript) arises from dislocations.

Figure R1: (a) DFXM strain map, with the black line indicating the profile used to map the strain decay from the dislocation. (b) The strain decay from the dislocation into the matrix, fitted to a $1/r$ model. (c) Sample-space projections of DFXM simulations for a phantom wall of edge dislocations in aluminum at varying tilt angles. The core of the central dislocation is marked with a red dot. The top three images utilize wavefront propagation, while the lower images use geometrical optics, as demonstrated by Poulsen et al. (JAC, 2021) and adapted from Borgi et al. (JAC, 2024). (d) Forward simulation of a single edge dislocation using geometrical optics. (e) Experimental validation of an extensive dislocation boundary network using weak beam contrast in annealed aluminum (Yildirim et al., 2023, Sci. Rep.).

We have modified the text to clarify these points on the observed dislocation contrast at the tip of the dendrite and its surroundings.

2. Even if the observed defect is dislocation, its role one dendrite growth may not be significant, neither to the electrochemical performance. Therefore its impact to the battery community is limited.

While we appreciate the reviewer's perspective, we believe our study provides significant insights into the growing behavior of dendrites in ceramic electrolytes. This research is highly relevant for researchers focused on the fundamental understanding of battery failure. Our findings align with and contribute to a body of work that has been published in reputable, high-impact journals. To name a few:

- For example, Ning et al. recently published a paper in Nature discussing dendrite growth in LPSCI-based electrolytes. They propose that Li metal does not completely fill flaws before fracture initiation, which has been well-received by the community despite the potential differences in practical applications for large-scale batteries. You can find their study via this link <https://www.nature.com/articles/s41563-021-00967-8>
- Additionally, Berger and Janek published a paper in Nature Communications on the evolution of lithium dendrites at $\text{Li}_{6.25}\text{Al}_{0.25}\text{La}_3\text{Zr}_2\text{O}_{12}$ grain boundaries using operando microscopy techniques. This work has also garnered significant attention and further illustrates the importance of understanding dendrite behavior. The paper is available in this link <https://www.nature.com/articles/s41467-023-36792-7>

We believe our study complements these contributions and adds valuable knowledge to the ongoing efforts in understanding and improving battery performance.

3. There is lack of experimental details, e.g. what was the experimental set-up for the dendrite growth? How was the electrochemical experiment was done? What sample was used for the study?

Thank you for your comment. We realize the methods section lacked clarity and have updated it to provide a more detailed explanation of the experimental setup. Additionally, we have included an overview of the setup in the supplementary information to offer a visual aid for the reader. The captions have also been updated to clearly indicate which sample was used for DFXM.

Reviewer #3:

The article titled "Origin of dendrite branching in solid-state batteries" by Yildirim and Flatscher et al. present findings on the interplay between the formation of dislocation and propagation of lithium dendrite in single crystal garnet-type LLZTO solid electrolyte using Dark-field X-ray Microscopy (DFXM). According to the presented results, dislocations are formed within the solid electrolyte due to the build-up of hydrostatic pressure caused by electric-field driven deposition of lithium at a dendrite. The dislocations formed in the material create a weakness, leading to preferential cracking of the LLZTO and resulting in branching of the dendrite structure.

The manuscript raises interesting claims and discussions about the mechanistic understanding of dendrite propagation within single crystal LLZTO, which are highly valuable for the solid-state battery community. Additionally, the visualization of dislocations and their correlation with dendritic structures is a significant breakthrough going beyond the current state of research. However, the conclusions drawn base only on a single post-mortem experiment, leaving much space for miss interpretation. As no initial DFXM was captured prior dendrite formation, it remains ambiguous whether dendrites propagation initiates dislocation formation or dendrites propagate already existing dislocation within the LLZTO.

Thank you for your valuable comments. We appreciate the opportunity to address the concern regarding the interpretation of our conclusions based on a single post-mortem experiment. While

it is true that our study is based on post-mortem analysis, we have included a new figure in Figure 2, showing a larger field of view around the dendrite.

The dislocation density near the dendrite tip (shown in Fig.2e-h) is $4.42 \times 10^8 \text{ m}^{-2}$ determined from ~12 dislocations in the area of $102 \times 246 \text{ }\mu\text{m}^2$. This is 2 orders of magnitude higher than what we could expect from a pristine sample. Compared to the larger FOV image in figure 2i where the dislocation density was determined as $1.14 \times 10^8 \text{ m}^{-2}$ with ~20 dislocations in an area of $700 \times 250 \text{ }\mu\text{m}^2$. This reduction in the dislocation density when moving away from the dendrite supports the conclusion that the dendrite and the dislocations are indeed connected. It has to be noted that these calculations represent an underestimation for the dislocation density as there may be more dislocations in this region that could not be detected from the probed reflections.

Furthermore TEM characterization shown in the supplementary information did not find dislocations though it is likely that a pristine area of the sample was probed, as it is by no means trivial to accurately cut and extract a FIB lamella especially with the reduced visibility of the dendrite in the electron microscope compared to the optical image. Given the low dislocation density in pristine regions, the probability that a dendrite tip would encounter a dislocation precisely at the moment we stopped the experiment is exceedingly low. This supports our observation that dislocations are related to dendrite formation rather than pre-existing dislocations within LLZTO.

We have modified the discussion section as follows:

“The typically low dislocation densities in single-crystalline ceramics, as minimal as $1 \times 10^6 \text{ m}^{-2}$, arise from uniform temperature control and extended periods at high temperatures that facilitate diffusion during processing. The calculated dislocation density around the dendrite tip, shown in Fig. 2(e-h) is approximately $4.42 \times 10^8 \text{ m}^{-2}$ with ~12 dislocations in the $102 \times 246 \text{ }\mu\text{m}^2$ area. Fig.2(i) which shows a lower magnification of the dendrite demonstrates a lower dislocation density of approximately $1.14 \times 10^8 \text{ m}^{-2}$ with ~20 dislocations populating the $250 \times 700 \text{ }\mu\text{m}^2$ area. Some of them are highlighted with black arrows in Fig.2(i). It has to be noted that these calculations represent an underestimation for the dislocation density as there may be more dislocations in this region that could not be detected from the probed reflections. The regions farther from the dendrite, where dislocations are sparse, can be considered closer to a pristine single crystal. It has to be noted that these calculations represent an underestimation for the dislocation density as there may be more dislocations in this region that could not be detected from the probed reflections. The regions farther from the dendrite, where dislocations are sparse, can be considered closer to a pristine single crystal. TEM characterization (Supplementary Figure 1) showed no observation of dislocations, though it is likely that a pristine region was observed as the reduced visibility of the dendrite in the electron microscope compared to the optical image causes accurate positioning of the FIB lamella to be a non trivial problem. Given the low dislocation density in these pristine regions, the probability that a dendrite tip would encounter a dislocation precisely at the moment the experiment was stopped is exceedingly low.”

Therefore, the manuscript is recommended for publication after addressing the mentioned concern. Additionally, the following points require further explanation or correction:

a. The actual title is misleading. The manuscript focuses within the introduction as well as the rest of the paper on Li batteries. I.e., no literature about solid electrolytes for Na batteries is taken into account. Furthermore, no real battery was assembled for the electrochemical experiments (only half cell). So, it might be fair to adapt the title to "Origin of dendrite branching in Li solid-state battery materials". – Or alternatively expand to other types of solid state batteries.

Reviewer's point is well made and definitely appreciated. We have changed the title to "The origin of dendrite branching in inorganic solid electrolytes"

b. While DFXM has been used in material science, the method is rather new in the field of solid-state batteries. It is necessary to explain how dislocation can be visualized by this method. Moreover, are the observed dislocations single dislocations or an average of several dislocations within a small voxel?

A similar question was asked by Reviewer #2, and we have already provided a detailed response. Please refer to our response to Reviewer #2's first question. In this study, the observed dislocations are individual dislocations characterized by the specific misorientations surrounding them. Unlike low-dislocation-density systems such as ceramics (Simons et al., ACS Nano 2019; Porz et al. Materials Horizons, 2021), semiconductors, or well-annealed metals (Dresselhaus-Marais et al. Science Advances 2021), DFXM effectively captures dislocation boundaries in higher dislocation density systems. This is demonstrated by the finite misorientation observed across dislocation boundaries, as illustrated in Yildirim et al. (2022 and 2023, Scripta Materialia), and Zelenika et al. (2024, Acta Materialia).

c. Why does the sample orientation in Figure 2e vary significantly along the highlighted dendrite starting from the tip inside the "bulk" of the Li?

The symmetric variation in the sample orientation in the dendrite vicinity stems from mechanical stress induced by the dendrite growth. As the dendrite penetrates the crystal, it effectively "opens" the crystal, resulting in a curvature in the crystal lattice that exhibits symmetry with a mirror plane defined by the dendrite itself.

d. Why is the sample orientation in Figure 2g opposite to the sample orientation given in Figure 2e?

Thank you for your observation regarding the sample orientation in Figures 2g and 2e. As shown in the experimental schematics in Figure 1, we used DFXM to measure local orientation fluctuations around the 532 diffraction vector. This involved rotating the sample in the "rock" and "roll" directions with the two sample tilts denoted as ϕ and χ . These two scans show different lattice distortions in perpendicular directions. The color coding is based on setting the zero value at the highest intensity point along the scan, corresponding to the peak position of the rocking curve.

e. Is it possible to further categorize the observed dislocation in screw or edge dislocation? If so, what type have been observed at the tip of the dendrite as well as on the branching parts?

The referee's question is indeed well posed and critical to the field. Our study focuses on characterizing the dislocation lines but does not explicitly address the type or Burgers vectors of the dislocations

To determine the type of dislocation, one should examine the contrast behavior using different g vectors and apply the invisibility criterion. In our experiment, we only measured the local strain and orientation fluctuations around the 532 diffraction vector. Our work on experimentally identifying dislocations is ongoing. We have projects with our collaborators to explore the invisibility criteria further and incorporate them into our characterization, but this is beyond the scope of this initial work.

f. Did you also observed dislocations in the sample investigated which are not located near the dendrite to support the statement of low dislocation density?

Yes, we did observe regions in the investigated sample where dislocations were not located near the dendrite. This is shown in the added Figure 2.(i) which shows a larger field of view map. There the dislocation density is lower with $1.12 \times 10^8 \text{ m}^{-2}$ compared to the dendrite tip with $4.42 \times 10^8 \text{ m}^{-2}$, though part of the dendrite is present in the middle of this region. It shows that the dislocation density is lower in areas distant from the dendrite, reinforcing our conclusion that the dislocations are predominantly associated with the dendrite formation rather than being uniformly distributed throughout the sample. Furthermore TEM characterization did not detect dislocations though there should have been some if the dislocation density in the range of $1 \times 10^8 \text{ m}^{-2}$ continued throughout the sample.

g. Since the manuscript bases on one post-mortem analysis only, a reference sample without dendrites should be measured in comparison and data given in the SI. If there are dendrite free areas within the investigated sample, these can also be taken as a reference.

Thank you for your comment. While we agree that a reference sample without dendrites would provide valuable comparison data, the new higher field of view figure included in Figure 2 (i) can be considered as representing pristine areas away from the dendrite. This figure covers an area of $700 \times 250 \times 0.6 \text{ } \mu\text{m}^3$ and demonstrates that regions distant from the dendrite exhibit fewer dislocations, as it transitions more to the pristine regions of the sample. This data supports our conclusions and provides a meaningful reference within the investigated sample.

h. Why does dendrite growth from right to left in Figure 3j, when the electrode is placed on top and bottom?

Thank you for bringing this to our attention. Although Figure 3j might suggest that the dendrite grows from right to left, it actually grows along the right edge and branches into the center as it develops. We have updated the caption to clarify this.

i. The authors suggest to increase the fracture toughness by increasing the dislocation density of the LLZTO as dislocation movement allows plastic deformation and thus pressure release. However, as outlined in the discussion the dislocation movement in ceramic is comparable low. So, would this be a good solution with respect to the kinetic time scale?

Thank you for this interesting question. While it is indeed true that the motion of dislocations in ceramics is lower than observed growth speeds of dendrites, growing dendrites necessitate a fracture of the solid electrolyte. If the plastic deformation can dissipate the stress before it reaches the level that is needed for fracture, then it should be a viable solution to extend the lifespan of the solid electrolyte at a given current density. Of course increasing the current density further may tip the balance back into favoring fracture but further research is going to be needed to give a more conclusive answer.

j. Statement of the “perfect single crystal” in given in caption of Figure 1 of the SI should be reevaluated, as TEM only gives a local point of view.

We have modified the text as follows:
“... the extracted lamella, showing that the sample is a near-perfect single crystal within the limitations of TEM technique.”

Additionally smaller comments:

a. Please check the chronological order in the first paragraph of the introduction. Maybe start to address liquid electrolytes and move to solid electrolytes. - The phrase “Similar challenges” does not fit in sense of the previous and following sentence.

We have modified the text as follows:

“Dendrites present a significant challenge in the development of high-energy Li-ion batteries. To combat dendrites, a deeper understanding of the causes for their nucleation and propagation is crucial. During electrodeposition, uneven metal deposits can lead to dendritic structures with higher local electric fields. This creates a cycle where metal preferentially accumulates in the high-field region, causing dendrite growth^{1,2}. In liquid electrolyte lithium-ion batteries, similar challenges arise. To mitigate dendrite growth, extensive research has focused on the utilization of additives and diverse charging protocols, aiming to achieve a more homogeneous lithium deposition²⁻⁶. Solid electrolytes, on the other hand, were postulated to stop dendrite growth by creating a mechanical barrier, where the growing soft lithium metal would be halted⁷. Their non-flammability would also eliminate safety issues⁸. However, studies have shown that soft lithium metal can still penetrate solid electrolytes, including sulfur-based materials such as Li₆PS₅Cl and mechanically robust oxide-based materials like the garnet Li₇La₃Zr₂O₁₂ (LLZO)⁹⁻¹¹. Given the theoretically strong mechanical properties of LLZO, this work primarily focuses on this material^{7, 12}.”

b. Please refer in the main text to the figures shown in the Supplementary Information. The supplementary figures are now referred to in the main text.

c. The term “dendrite formation” is too general as it include the initial nucleation as well as the propagation process. It’s suggested to use more specific terms to describe the observed phenomena, like “dendrite initiation” and “dendrite propagation”.

Dendrite formation has been updated to dendrite propagation in the text.

d. Please highlight the dendrite structure in the Figure 2b, c and d to help the reader identifying the mentioned “certain local orientation of LLZTO” and the surrounding. The dendrite structure has been highlighted in Figure 2b-d with dotted lines.

e. We observed issues with opening supplementary video 2, as it is coded with png. data type. Thank you for mentioning this issue. We have modified the video format and also corrected the aspect ratio.

f. P. 5: “in-operando measurement” is not existing. In contrast to in-situ the right wording is “operando measurement”.

The wording has been modified accordingly.

T. Ortmann and M. Rohnke

Reviewer #4:

I co-reviewed this manuscript with one of the reviewers who provided the listed reports. This is part of the Nature Communications initiative to facilitate training in peer review and to provide appropriate recognition for Early Career Researchers who co-review manuscripts

REVIEWERS' COMMENTS

Reviewer #1 (Remarks to the Author):

The authors have addressed all my concern, and I am happy to see its publishing in NC.

Reviewer #2 (Remarks to the Author):

The authors have addressed all my questions, I recommend publication.

Reviewer #3 (Remarks to the Author):

The authors carefully revised the manuscript. All open questions were addressed and well answered. From my point of view the manuscript can be published as it actually is. I recommend the publication in Nature Communications.

Reviewer #4 (Remarks to the Author):
